# Oral-Health-Related Quality of Life in Patients with Medication-Related Osteonecrosis of the Jaw: A Prospective Clinical Study

**DOI:** 10.3390/ijerph191811709

**Published:** 2022-09-16

**Authors:** Anna Winter, Stefan M. Schulz, Marc Schmitter, Roman C. Brands, Anton Straub, Alexander Kübler, Anna Borgmann, Stefan Hartmann

**Affiliations:** 1Department of Prosthodontics, Julius Maximilian University Würzburg, Pleicherwall 2, 97070 Würzburg, Germany; 2Department of Behavioural Medicine and Principles of Human Biology for the Health Sciences, Trier University, Universitätsring 15, 54296 Trier, Germany; 3Department of Oral and Maxillofacial Plastic Surgery, University Hospital Würzburg, Pleicherwall 2, 97070 Würzburg, Germany

**Keywords:** oral-health-related quality of life, satisfaction with life, oral health, medication-related osteonecrosis of the jaw, treatment benefit, OHIP-49, SWLS

## Abstract

Medication-related osteonecrosis of the jaw (MRONJ) represents an adverse side effect of antiresorptive and antiangiogenic medications. It is associated with impaired quality of life, oral health, and oral function and can be classified into various stages. The purpose of this prospective clinical study is to evaluate the impact of stages I and II MRONJ on oral-health-related quality of life (OHRQoL) and related parameters. Patients’ OHRQoL, satisfaction with life, oral discomfort, and oral health were assessed using the German version of the Oral Health Impact Profile (OHIP-G49), visual analog scales (VAS), and Satisfaction with Life Scale (SWLS) at baseline (T0), 10 days (T1), and 3 months after treatment (T2) in 36 patients. Data were analyzed using Kolmogorov–Smirnov test, two-way mixed ANOVAs, and follow-up Mann–Whitney U tests. The impact of treatment effects on the original seven OHIP domain structures and the recently introduced four-dimensional OHIP structure were evaluated using linear regression analysis. Thirty-six patients received surgical MRONJ treatment. Before treatment, patients’ perceived OHRQoL, oral discomfort, oral health, and satisfaction with life were negatively affected by MRONJ. Surgical treatment significantly improved OHRQoL and related parameters (all *p* ≤ 0.012). This improvement was greater in patients with higher impairment at T0. OHRQoL and oral restrictions were still impaired after treatment in patients who needed prosthetic treatment. The four-dimensional structure revealed valuable information beyond the standard seven OHIP domains. Increased awareness of MRONJ risks and an interdisciplinary treatment approach for MRONJ patients are needed.

## 1. Introduction

Bone-modifying agents such as bisphosphonates or monoclonal antibodies are commonly used to prevent or treat bone-related diseases such as osteoporosis, bone metastases of solid tumors, and cancerous bone lesions of multiple myeloma [1,2]. These agents influence bone remodeling by modifying the differentiation and function of osteoclasts in different ways [3]. Bisphosphonates such as zoledronic acid or alendronat inhibit osteoclast-mediated bone resorption by inhibiting the mevalonate pathway [3]. Denosumab is a monoclonal antibody against the receptor activator of nuclear factor-kappa B ligand (RANKL). It reduces the enhanced osteoclast activity caused by tumor cells [4] and can prevent hypercalcemia and bone destruction and the production of tumor-producing growth factors [5]. Jakob et al. described an increased occurrence of MRONJ after the treatment with denosumab [6].

Despite their benefits, antiresorptive and antiangiogenic medications such as bevacizumab and sunitinib have been associated with an adverse side effect named medication-related osteonecrosis of the jaw (MRONJ) [7,8,9]. The American Association of Oral and Maxillofacial Surgeons (AAOMS) has defined MRONJ according to the following conditions: “current or previous treatment with antiresorptive or antiangiogenic agents; exposed bone or bone that can be probed through an intraoral or extraoral fistula in the maxillofacial region that has persisted for longer than 8 weeks; and no history of radiation therapy to the jaws or obvious metastatic disease to the jaws” [8].

The pathogenesis of MRONJ has not been fully elucidated [10], but some mechanisms have been hypothesized. It is hypothesized that, in case of inflammation, bone can be exposed in the oral cavity due to osteoclast inhibition and missing bone resorption after MRONJ-associated drug intake [11,12]. Subsequent bacteria colonizing by the microbial flora containing bacterial toxins of the oral cavity can lead to osteonecrosis of the jaw and MRONJ, respectively [11,12]. Others have suggested that antiresorptive drugs can weaken the microarchitectural structure of bone, which may lead to MRONJ [9].

The development of MRONJ depends on various risk factors. Systematic risk factors include chronic disease, age, smoking, or alcohol intake, and local risk factors such as anatomy and genetic factors have also been described [13,14]. MRONJ can occur spontaneously or following triggers such as invasive dental procedures such as dental extractions or treatments that expose and manipulate the bone [15]. Periodontal diseases or poorly fitting dentures can also lead to MRONJ [7].

To classify the progression of MRONJ, the AAOMS has defined five stages (Table 1). These stages are based on medication intake, clinical symptoms, and radiographic findings and affect how the MRONJ should be treated [8,13]. In general, pharmacological and surgical treatment methods such as sequestrectomy, debridement, and resection represent standardized treatment methods [16]. During the last decade, there was a drift towards a more aggressive surgical approach since watchful waiting strategies oftentimes failed [17].

Previous studies have reported a predominance of stages I and II MRONJ [16,18]. The clinical signs of stage I and stage II MRONJ can affect oral functions such as chewing, smiling, and speaking and can influence oral health [19,20]. Moreover, quality of life is negatively affected by MRONJ [21]. In order to assess the patients’ perspective of their diagnosis and treatment, dental patient-reported outcomes (dPROs) have been used. The Oral Health Impact Profile (OHIP) is a reliable and widely used questionnaire that assesses the patient’s perceived oral-health-related quality of life (OHRQoL) [22]. The questionnaire has summarized the OHIP in seven domains as standard, but an additional four-dimensional structure has recently been introduced [23].

To improve the care and outcomes of patients with MRONJ, a deeper understanding of how MRONJ affects OHRQoL is necessary. This increased knowledge would also better inform clinicians and, in turn, patients about treatment outcomes and potential effects of MRONJ on OHRQoL. This is important because patients have a fundamental right to this information. Moreover, providing information on how MRONJ affects OHRQoL-related parameters could increase the awareness of patients “at risk”, and the importance of prevention increases in general [24].

The aim of this study was to investigate the impact of stages I and II MRONJ and its associated parameters and treatment on OHRQoL outcomes. In addition, the novel four-dimensional structure of the OHIP was compared with the standard seven-domain structure in MRONJ patients. The following null hypotheses were stated:(I)Stages I and II MRONJ and its successful therapy predict no improvement of OHRQoL(II)MRONJ-associated parameters do not affect OHRQoL.

## 2. Materials and Methods

### 2.1. Study Population

The present clinical trial was conducted from November 2020 until January 2022 in the Department of Oral and Maxillofacial Plastic Surgery at the university hospital of Würzburg in Germany. Patients undergoing surgical treatment for stage I or stage II MRONJ were consecutively recruited according to the classification of the AAOMS. After a detailed explanation of the study, written informed consent was obtained from all participants. Inclusion criteria were at least 18 years of age and a diagnosis of stages I or II MRONJ. Exclusion criteria were not given consent to participate, being unable to complete the questionnaires, such as for patients with cognitive impairments. The study was approved by the local ethics committee of the University School of Medicine (approval number: 139/18).

### 2.2. Data Collection

All patients were examined by a specialist in oral and maxillofacial plastic surgery at their first appointment before MRONJ treatment (T0). Descriptive parameters and anamnestic parameters related to MRONJ were recorded by the physician, and these parameters were used to define the subgroups as follows:(I)Total: included the entire patient sample(II)Stage: clinical stages I and II(III)Pain: presence (yes) or absence (no) of pain(IV)Primary disease: the disease for which MRONJ-related drugs were used. These were divided into two groups: osteoporosis and malignoma. Malignoma included multiple myeloma and breast, lung, prostate, renal cell, thyroid, and gastric carcinomas.(V)Risk evaluation: risk was evaluated according to drug intake. A low-risk group (Prolia/denosumab, 60 mg, subcutaneous administration every 6 months and alendronate, 70 mg, oral application once a week) was compared with a high-risk group (Xgeva/denosumab, 120 mg, subcutaneous application every 4 weeks and zoledronate, 4 mg, intravenous administration every 4 weeks)(VI)Duration of intake: was categorized into short (≤36 months) and long (>36 months) duration of drug intake(VII)Defect size: MRONJ-related defects were categorized into small (<2 cm^2^), medium (≥2 to <4 cm^2^), and large (≥4 cm^2^) defects(VIII)Need for prosthodontics: Patients who needed prosthetic care (yes) were compared with patients who did not need prosthetic treatment (no).

Patients were examined again after MRONJ treatment at T1 and T2 appointments. All examinations were performed by one operator.

### 2.3. Questionnaires

All patients completed the questionnaires at baseline before MRONJ treatment (T0), approximately 10 days after treatment (T1), and 3 months after treatment (T2).

The first part of the questionnaire contained the German version of the Oral Health Impact Profile 49 (OHIP-G49), which is used as standard to assess OHRQoL and treatment effects [25]. The OHIP-G49 questionnaire asks how often patients experienced a given symptom in the previous week and covers seven domains: functional limitation, physical pain, psychological discomfort, physical disability, social disability, psychological disability, and handicap [22]. Recently, John et al. classified the OHRQoL into four-dimensions: oral function, orofacial pain, orofacial appearance, and psychosocial impact [23]. Each Item was assessed on a five-point Likert scale (0 = never to 4 = very often) [22], with a lower OHIP score indicating a better evaluation of OHRQoL. A total OHIP score was calculated by adding the scores of all items together. Scores for each of the domains and dimensions were calculated by adding the scores of the item related to the dimension.

In order to obtain more information on OHIP, patients were asked about their current oral function and status using a visual analog scale (VAS) as in previous studies [26,27]. Patients were asked to what extent they felt restricted by their oral status because of oral discomfort (VAS1). Patients’ perceived oral health and function were assessed in questions 2 to 6 (VAS2-6) by their ability to chew, swallow, and taste, which was summarized for statistical analysis. Patients marked their answers on the scale; the left end of the scale represented the minimum extension of the item (=0), and the right end represented the maximum extension of the item (=100). A higher score indicated a more negative evaluation by the patient. Finally, to prove the comprehensibility and rationale of the questions, patients were interviewed after the assessment at T0.

Satisfaction with life was assessed using the German version of the Satisfaction With Life Scale (SWLS). The SWLS asks respondents to score five items related to general life satisfaction on a seven-point Likert scale (1 = strongly disagree, 7 = strongly agree) [28]. A higher sum score indicates higher life satisfaction.

### 2.4. Statistical Analysis

The Kolmogorov–Smirnov test was used to test for normal distribution, and the two-way mixed ANOVA was used to determine the main effects of assessment time, group, and interaction between time and group. Mauchley’s test was used to test for sphericity, and if sphericity was lacking, Greenhouse–Geisser correction was used. The Follow-up Mann–Whitney U test was used to examine differences between the groups at specific times. Changes (improved, deteriorated, or no change) in stage and defect size were assessed, and their effects on OHIP domains in the standard seven-domain questionnaire and the new four-dimensional structure were determined using linear regression models. Missing data (<1%) were imputed using the last observation carried forward approach. Partial eta squared was added as a measure of effect size, with 0.01, 0.06, and 0.14 reflecting small, medium, and large effects, respectively. Cronbach’s alpha was used in order to evaluate internal consistency. Data were analyzed with SPSS version 28.0 (SPSS, Chicago, IL, USA), and the default level of significance was set at α ≤ 0.05. In order to account for multiple testing, Bonferroni correction was applied, resulting in an α ≤ 0.0125 for the ANOVA results.

## 3. Results

### 3.1. Patient Characteristics

Thirty-six patients (13 males and 23 females) were included. MRONJ was caused by a tooth extraction in all patients. The mean age of all patients was 68.8 (SD = 14.0) years. MRONJ-related parameters are described in Table 2.

Additional subgroups were formed according to the change of stage between T0 and T1: 14 patients improved, 1 patient deteriorated, and 21 patients showed no change. Similar results were observed in change of stage between T0 and T2: 13 patients improved, one patient deteriorated, and 22 patients showed no change. Regarding change of defect size between T0 and T1, an improvement was found in 22 patients, deterioration in two patients, and no change in 12 patients. For change of defect size between T0 and T2, 21 patients improved, two patients deteriorated, and 13 patients showed no change.

### 3.2. OHIP Evaluations

In general, OHIP scores decreased between T0 and T1/T2 (Figure 1). Evaluation of the OHIP sum score showed a significant main effect for time (T0/T1/T2) for all parameters (stage, pain, primary disease, risk evaluation, duration of drug intake, defect size, and need for prosthodontics) (all F [2,68] ≥ 8.558, all *p* ≤ 0.001, partial-η^2^ [min to max] = 0.201 to 0.322). Improvement occurred between T0 and T1 (all F [1,34] ≥ 7.756, all *p* ≤ 0.009, partial-η^2^ [min to max] = 0.186 to 0.355). This improvement was maintained at T2 but no further significant improvements were observed between T1 and T2 (all F [1,34] < 0.858, all *p* > 0.213, partial-η^2^ [min to max] = 0.025 to 0.045).

No significant main effects between groups were found (all F [1,34] < 0.003, all *p* > 0.05, partial-η^2^ [min to max] = 0.008 to 0.108). Although no significant differences were observed in the need for prosthodontics between groups, an almost large effect size was observed within the group (partial-η^2^ = 0.101).

Statistically significant time by group interactions were found for risk evaluation (F [2,68] = 5.960, *p* ≤ 0.004, partial-η^2^ = 0.149). Planned contrasts revealed that the change in risk evaluation was significantly different in all groups between T0 and T1/T2 (all F [1,34] ≥ 7.267, all *p* ≤ 0.011, partial-η^2^ [min to max] = 0.176 to 0.206), but not between T1 and T2 (F [1,34] = 0.019, *p* > 0.892, partial-η^2^ = 0.001). Duration of drug intake only changed significantly between T0 and T2 assessments and this difference was dependent on group (F [1,34] = 7.086, *p* ≤ 0.012, partial-η^2^ = 0.172). Descriptive data for OHIP scores are presented in Table 3.

The highest pretreatment scores were found in patients with the need for prosthodontics and in the low-risk medication group.

### 3.3. SWLS Assessment

SWLS sum scores showed no significant main effect for time across T0/T1/T2 evaluations (all F [2,68] < 0.317, all *p* ≥ 0.153, partial-η^2^ [min to max] = 0.018 to 0.054) (Figure 2). However, mean scores showed a tendency to increase from T0 to T1/T2 (Table 4).

No significant main effect was found between groups (all F [1,34] < 0.011, all *p* ≥ 0.060, partial-η^2^ [min to max] = 0.001 to 0.101), except for defect size (F [1,34] < 4.591, *p* ≤ 0.0017, partial-η^2^ = 0.218). Follow-up U-tests for the separate assessments revealed a significant difference between patients with medium and large defects at T1 (*p* ≤ 0.009). The differences between stages I and II showed a medium effect size (partial-η^2^ = 0.101), but were not statistically significant (*p* = 0.060). No statistically significant time by group interactions were found (all F [1,34] < 3.749, all *p* > 0.029, partial-η^2^ [min to max] = 0.003 to 0.099). Descriptive data for SWLS scores are presented in Table 4.

### 3.4. VAS 1

VAS1 values showed a significant main effect for time (T0, and T1/T2) in all parameters (Greenhouse–Geisser, all F [2,68] ≥ 7.845, all *p* ≤ 0.002, partial-η^2^ [min to max] = 0.187 to 0.288). VAS1 total score is demonstrated in Figure 3.

Significant changes were observed for pain, risk evaluation, duration of drug intake, defect size, need for prosthodontics, and dental status between T0 and T1, T0 and T2, and T1 and T2 (all F [1,34] ≥ 4.969, all *p* ≤ 0.012, all partial-η^2^ [min to max] = 0.128 to 0.390). With regard to the stage, values significantly improved from T0 to T2 (F [1,34] ≥ 13.374, *p* ≤ 0.001, partial-η^2^ [min to max] = 0.103 to 0.135) and in primary disease, values decreased significantly from T0 to T1/T2 (all F [1,34] ≥ 6.301, all *p* ≤ 0.001, partial-η^2^ = 0.282). VAS1 values showed no significant main effect between groups (all F [1,34] < 0.074, all *p* > 0.086, partial-η^2^ [min to max] = 0.002 to 0.073) and no statistically significant time by group interactions (Greenhouse–Geisser, all F [2,68] < 0.074, all *p* > 0.195, partial-η^2^ [min to max] = 0.002 to 0.073). Mean VAS1 values are shown in Table 5.

### 3.5. VAS2-6

VAS2-6 values showed a significant main effect for time from T0 to T1/T2 in all parameters (Greenhouse–Geisser, all F [2,68] ≥ 6.232, *p* ≤ 0.006, partial-η^2^ [min to max] = 0.155 to 0.203). VAS2-6 total score is demonstrated in Figure 4. Change across time was significant for all parameters from T0 to T2 (all F [1,34] ≥ 9.369, all *p* ≤ 0.004, partial-η^2^ [min to max] = 0.216 to 0.288), and from T1 to T2 (all F [1,34] ≥ 6.975, all *p* ≤ 0.012, partial-η^2^ [min to max] = 0.170 to 0.200).

No significant main effects were found between groups (all F [1,34] < 0.056, all *p* > 0.122, partial-η^2^ [min to max] = 0.018 to 0.096). Moreover, there was no statistically significant interaction between time and group (Greenhouse–Geisser, all F [2,68] < 0.042, all *p* > 0.299, partial-η^2^ [min to max] = 0.001 to 0.059). Mean VAS2-6 values are presented in Table 6.

### 3.6. Analysis of Linear Regression

Regression analysis of the original seven domain structure demonstrated a significant impact of change of stage between T0 and T1 on T2 evaluations of OHIP domains. An improvement in stage over time predicted a decrease in functional limitation (*p* ≤ 0.037, β = 0.349, R^2^ = 0.122) and psychological discomfort (*p* ≤ 0.039, β = 0.345, R^2^ = 0.119). Moreover, a reduction in defect size between T0 and T1 predicted a significant reduction in OHIP scores in the handicap domain at T1 (*p* ≤ 0.046, β = −0.334, R^2^ = 0.112).

Regression analysis of the four-dimensional structure showed no significant effects for oral function (*p* ≥ 0.267, β ≤ 0.190, R^2^ ≤ 0.036), orofacial appearance (*p* ≥ 0.128, β ≤ 0.259, R^2^ ≤ 0.067), and psychosocial impact (*p* ≥ 0.084, β ≤ 0.292, R^2^ ≤ 0.085). However, orofacial pain at T0 was significantly affected by the change of stage between T0 and T2 (*p* = 0.019, β = 0.388, R^2^ = 0.151) and change of defect size between T0 and T1 (*p* = 0.010, β = 0.422, R^2^ = 0.178) and T0 and T2 (*p* = 0.022, β = 0.380, R^2^ = 0.145). Moreover, change of stage between T0 and T1 (*p* = 0.014, β = 0.406, R^2^ = 0.165) and T0 and T2 (*p* = 0.010, β = 0.421, R^2^ = 0.178) significantly affected T2 evaluation of the dimension orofacial pain. Thus, an improvement of MRONJ stadium across time predicted a decrease in T2 evaluation. Full explorative data of OHIP domains according to the seven-domain questionnaire and four-dimensional structure are presented in Appendix A.

### 3.7. Internal Consistency

Cronbach’s alpha test demonstrated values between 0.770 (VAS 1/VAS2-6) and 0.965 (OHIP-49) at baseline assessment. Follow up assessments revealed values between 0.817 (VAS 1/VAS2-6) and 0.961 (OHIP-49) at T1, and 0.818 (VAS 1/VAS2-6) to 0.955 (OHIP-49) at T2.

## 4. Discussion

The aim of this study was to evaluate the impact of stages I and II MRONJ and its treatment on OHRQoL to better understand which impairments an MRONJ diagnosis and treatment have on patients. The results indicate that the diagnosis and treatment of MRONJ affect the OHRQoL of patients but mainly without significant differences between groups. Therefore, the first and partially second null hypotheses can be rejected.

MRONJ significantly worsened the OHRQoL before treatment in patients with malignoma and osteoporosis, and patients reported that this reduced OHRQoL affected them negatively. These findings are in line with the results of OHRQoL-specific questionnaires in other patient cohorts. For example, Miksad et al. observed significant impairment of OHRQoL due to bisphosphonate-associated osteonecrosis of the jaw [29]. This is in agreement with the findings of Caminha et al., who described severely restricted OHRQoL at the time of stages I and II MRONJ diagnosis in cancer patients [30]. De Cassia Tornier et al. also showed a significant impairment of OHRQoL due to MRONJ [31].

It was demonstrated that OHRQoL, oral function, and general satisfaction with life were improved in patients after successful treatment. This corroborates previous reports showing that surgical treatment improves OHRQoL in patients with stages I and II MRONJ [32,33]. These findings are also in line with those of previous studies showing that standard surgical interventions are more effective than conservative treatment [34,35].

Satisfaction with life, evaluated with the SWLS questionnaire, also improved after successful MRONJ treatment, but this effect was not significant. However, defect size did have a significant effect on satisfaction with life—patients with larger defects were significantly less satisfied with life at all assessments. To the best of our knowledge, the SWLS questionnaire has not been used to investigate the effects of MRONJ on patients, so we cannot compare these findings with those of previous studies. However, questions related to mental health in the Short-Form-Health-Survey-12 (SF-12) questionnaire give information on satisfaction with life [36]. Using the SF-12 questionnaire, Capocci et al. showed no significant correlation between satisfaction with life and MRONJ stage [37], which is in accordance with our findings. Moreover, the SWLS scores in our participants are comparable to those reported in the German population (26.38 ± 5.13 in males and 24.14 ± 5.94 in females) [28], suggesting that MRONJ does not directly impair satisfaction with life. We recommend the SWLS as an interesting tool for measuring satisfaction with life in future studies, particularly in larger samples.

No significant differences in the parameters we investigated were observed; however, we did note some clinically relevant tendencies. Before treatment, patients with stage I MRONJ had lower OHRQoL, less satisfaction with life, lower self-perceived oral health, and higher oral discomfort than patients with stage II MRONJ did. This contradicts the findings of previous studies, which described lower OHRQoL in patients with stage II MRONJ. However, these effects were also not statistically significant [33].

Oral function was investigated before and after treatment. OHIP scores three months after treatment were more improved in patients diagnosed with stage II MRONJ than in patients diagnosed with stage I MRONJ at baseline, which is in accordance with the findings of Sato et al. [33]. However, stage II patients experienced a less oral function and more oral discomfort and are still more impaired after therapy than stage I patients did. El-Rabbany et al. evaluated the ability to swallow, taste, and chew in patients with MRONJ and observed impairment in these oral functions before and still after MRONJ therapy [32]. Impairment in oral functions after treatment can be due to MRONJ-related pain, intraoral infections, or necrotic bone. Furthermore, taste can be impaired by oral malodor due to anaerobic procedures during osteonecrosis of the jaw [31,33,38]. The primary disease can also influence oral symptoms after MRONJ treatment. Furthermore, the taste alteration was greater and oral functions were more impaired in patients who received radiotherapy [39]. Taken together, these findings indicate that patients with stage II MRONJ do not recover as well in terms of oral health and OHRQoL as patients with stage I MRONJ do.

Although they were not significant, differences in OHIP scores between patients with and without the need for prosthodontic treatment had an almost large effect size. This underlines the lower OHRQoL in patients who need prosthodontic treatment before and after MRONJ therapy. This difference in OHIP scores is also reflected in the higher restriction of oral functions such as chewing and swallowing that we observed at baseline and follow-up assessments; however, these differences were not statistically significant. Oral functions are generally impaired in patients with MRONJ, as shown, for example, by Oteri et al., who also described higher impairment of oral health and more oral discomfort in patients with MRONJ [16,40].

We also found that the need for prosthodontics reduced the terms of OHRQoL. In a previous study, a patient sample that was similar in age to our sample and also needed prosthodontics but did not have MRONJ had an OHIP summary score of 41.2 ± 31.6, which is lower than in our patient groups [41]. In support of our findings, other studies have shown that the need for prosthodontics impairs the OHRQoL by reducing chewing ability, oral health, and aesthetics [41,42]. In this study, it was demonstrated that MRONJ combined with the need for prosthodontics reduced the OHRQoL at baseline and follow-up even further than MRONJ alone. Based on these findings, we recommend an interdisciplinary treatment approach to improve oral functions and OHRQoL after MRONJ treatment.

Moreover, the effect of defect size on OHRQoL was investigated, which is easy to evaluate and standardize. Patients with larger defects benefited significantly more from treatment than patients with smaller defects did. Tooth extraction was the cause of MRONJ in all our participants and is a known risk factor for MRONJ [43]. The high prevalence of large defects in our study may be explained by risk factors, such as high-risk medication (78%), long-duration intake (65%), and malignoma as the primary disease (100%). The defect size was reduced by treatment in 86% of patients with large defects. Improvement was lower in patients with small (<2 cm^2^, 77% improvement) and medium (≥2 to <4 cm^2^, 33% improvement) defects. In addition, defect size increased over time. This could explain the major improvement we observed in OHRQoL in patients with large defects and the lower satisfaction with life observed in these patients.

Changes in stage and defect size also had a significant effect on OHIP domains. An improvement in stage and a decrease in defect size were associated with improved scores in several of the seven OHIP domains. This supports the validity and sensitivity of the measure. Specifically, an improvement in stage predicted less functional limitation and reduced psychological discomfort. A decrease in defect size reduced OHIP scores in the handicap domain. In the four-dimensional scale, change in stage could only predict orofacial pain. Interestingly, we observed that the seven OHIP domains were more sensitive to short-term effects (i.e., those measured at T1), whereas the four-dimensional scale detected long-term differences in orofacial pain at T2. However, both OHIP scales showed the effects of MRONJ therapy at follow-up assessments, suggesting that both structures can be useful in assessing patients with MRONJ, which provides different aspects of interesting information. To the best of the author’s knowledge, this was not investigated before, and the use of the four-dimensional scale can be recommended in addition to the already established seven-domain scale in MRONJ patients, therefore.

There are several limitations to our study. First, only a small number of patients were included because of the low incidence of MRONJ [44]. However, our sample is comparable in size to those of previous studies evaluating the impact of MRONJ on quality of life in 20, 30, and 41 patients [21,37,40]. The subgroups contained varying numbers of patients, so missing significant differences within groups could be due to low test power. Nevertheless, we believe that the results from our subgroup analyses make a valuable theoretical and clinical contribution to the field, and we hope they will inspire further studies with larger and more uniform subgroups. However, the reliability of significant ANOVA results was supported by large effect size and linear regression results. In addition, the usage of various questionnaires enabled the assessment of different physical and mental conditions from the patient’s perspective. Moreover, the internal consistency of the questionnaires was good, and Cronbach’s alpha values were comparable to previous studies indicating good reliability and ability to measure OHRQoL and related terms [26,28,45].

Finally, we did not characterize the need for prosthodontics or the treatment, which may have affected the OHRQoL. For example, previous research has shown that the number of missing teeth and the type of surgical treatment (sequestrectomy, debridement, resection) can affect oral health and OHRQoL, therefore [8,13]. However, subgroup analyses of these parameters would have been unreliable because of the limited patient numbers, and further studies with higher patient numbers were recommended.

## 5. Conclusions

Within the limitations of the study, the following conclusions can be drawn:Considering the impaired OHRQoL due to MRONJ, awareness of MRONJ in dentists and patients should be increased, in order to improve patient compliance, especially in the MRONJ stage “at risk”.The present OHIP, SWLS, and VAS findings demonstrated that patients with stage II MRONJ, pain, and larger defects who have taken drugs for a longer time and need prosthetic treatment have greater changes in OHRQoL between baseline and follow-up. This could demonstrate a greater benefit of therapy in these patients. In general, the surgical treatment improved OHRQoL and related parameters. This underscores the benefit of surgical intervention, not only limited to effects on bone and mucosal healing but also in terms of quality of life.The impairment of MRONJ patients indicated that prevention, early diagnosis, early therapy, and interdisciplinary treatment might improve outcomes in patients with MRONJ, both in terms of their somatic health and subjective experiences such as OHRQoL.The effects of MRONJ treatment affected the seven-domain and four-dimensional scales. In addition, the new four-dimensional OHIP scale revealed additional information about the standard seven-domain structure.

## Figures and Tables

**Figure 1 ijerph-19-11709-f001:**
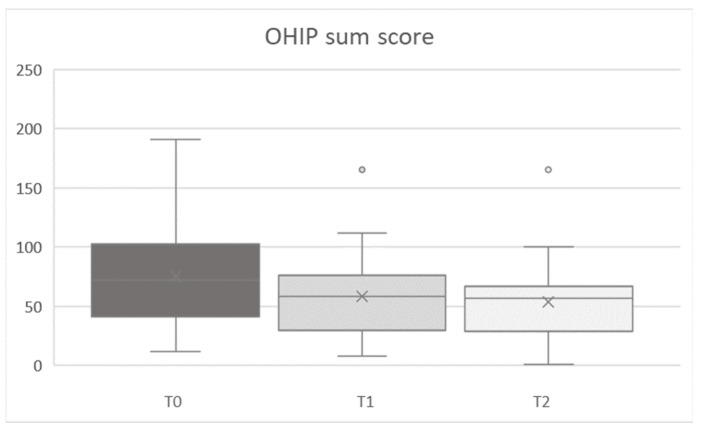
Total OHIP score at baseline (T0), first (T1) and second (T2) follow-up appointment.

**Figure 2 ijerph-19-11709-f002:**
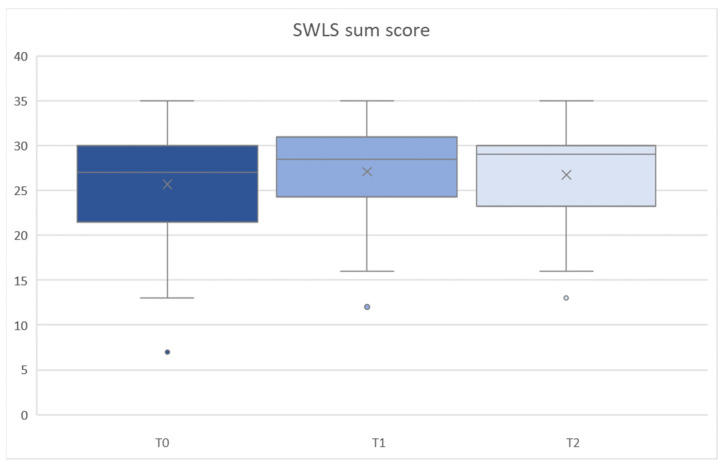
SWLS sum score at all assessment times (T0/T1/T2).

**Figure 3 ijerph-19-11709-f003:**
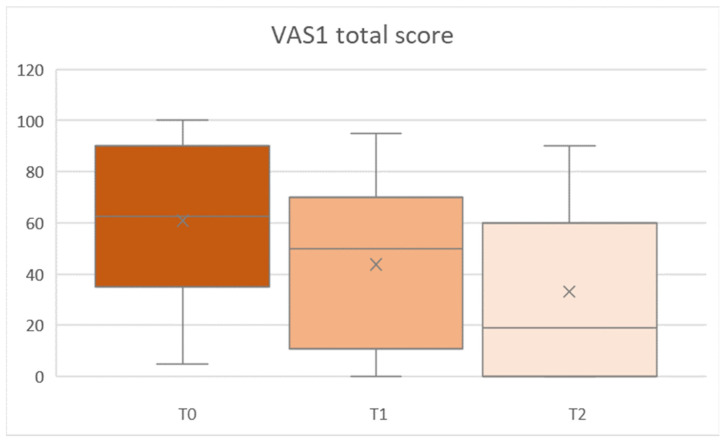
Total VAS1 score at baseline (T0), first (T1) and second (T2) follow-up assessments.

**Figure 4 ijerph-19-11709-f004:**
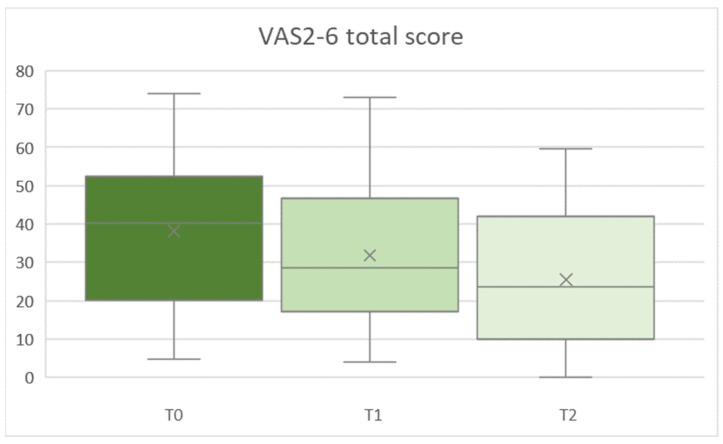
VAS2-6 sum score T0 (baseline, T1 (first follow-up) and T2 (second follow-up) assessment times.

**Table 1 ijerph-19-11709-t001:** MRONJ classification according to the American Association of Oral and Maxillofacial Surgeons.

Stage	Classification
At risk	patients who are being or have been treated with antiangiogenetic or antiresorptive therapy
Stage 0	prodromal symptoms without bone exposure
Stage I	bone necrotic and exposed, fistula, no signs of infection
Stage II	signs of infection additional to symptoms in stage I
Stage III	symptoms of stage II combined with pathologic fractures, extraoral fistula, or extensive osteolysis

**Table 2 ijerph-19-11709-t002:** MRONJ-related parameters in the different subgroups.

Parameter	Groups	*n*
Total	-	36
Stage	I	25
	II	11
Pain	no	15
	yes	21
Primary disease	osteoporosis	8
	malignoma	28
Risk evaluation	low risk	12
	high risk	24
Duration of intake	short	15
	long	21
Defect size	small	11
	medium	11
	large	14
Need for prosthodontics	yes	17
	no	19

**Table 3 ijerph-19-11709-t003:** OHIP scores for all parameters investigated and times of assessment (T0/T1/T2). SD: standard deviation.

Time of Assessment		T0		T1		T2	
Parameter	Groups	Mean	SD	Mean	SD	Mean	SD
Total	-	75.4	40.5	58.1	33.1	53.6	32.4
Stage	I	78.4	38.8	59.9	32.7	57.3	34.0
	II	68.6	43.2	54.1	35.2	45.3	28.1
Pain	no	67.6	29.6	53.8	29.1	48.5	26.7
	yes	81.0	46.7	61.2	36.1	57.3	36.1
Primary disease	osteoporosis	96.6	30.1	61.9	23.7	53.8	32.3
	malignoma	69.3	41.5	57.0	35.6	53.6	33.0
Risk evaluation	low risk	92.5	28.3	54.8	23.8	51.1	30.5
	high risk	66.8	43.4	59.9	37.3	54.9	33.8
Duration of intake	short	61.6	29.8	56.3	31.9	56.1	16.5
	long	85.2	44.8	59.4	34.7	51.9	40.4
Defect size	small	77.3	33.1	67.1	23.8	63.6	24.5
	medium	76.9	46.6	62.0	42.3	50.0	43.2
	large	72.7	43.5	48.0	30.8	48.7	28.3
Need for prostho- dontics	yes	89.9	41.1	67.7	36.2	61.3	36.4
	no	62.4	36.2	49.6	28.4	46.8	27.5

**Table 4 ijerph-19-11709-t004:** SWLS sum score for all parameters investigated and times of assessment (T0/T1/T2). SD: standard deviation.

Time of Assessment		T0		T1		T2	
Parameter	Groups	Mean	SD	Mean	SD	Mean	SD
Total	-	25.7	6.4	27.1	6.2	26.7	5.5
Stage	I	25.0	6.3	25.7	6.7	25.6	5.9
	II	27.2	6.7	30.3	3.3	29.4	3.3
Pain	no	24.8	7.7	25.1	8.0	24.8	6.9
	yes	26.3	5.4	28.6	4.0	28.1	3.9
Primary disease	osteoporosis	26.1	5.9	23.3	8.3	26.1	5.0
	malignoma	25.6	6.6	28.2	5.1	26.9	5.7
Risk evaluation	low risk	26.7	6.3	26.3	8.2	27.3	4.9
	high risk	25.2	6.6	27.5	5.0	26.5	5.9
Duration of intake	short	25.5	7.4	27.8	5.8	27.0	5.7
	long	25.8	5.8	26.6	6.5	26.5	5.5
Defect size	small	26.9	6.2	28.0	5.6	27.9	4.4
	medium	22.1	7.5	23.4	6.5	23.5	5.7
	large	27.6	4.6	29.4	5.3	28.4	5.4
Need for prostho-dontics	yes	25.7	5.4	26.5	5.7	25.8	6.1
	no	25.7	7.3	27.6	6.7	27.6	4.9

**Table 5 ijerph-19-11709-t005:** VAS1 values for all parameters and assessment times (T0/T1/T2). SD: standard deviation.

Time of Assessment		T0		T1		T2	
Parameter	Groups	Mean	SD	Mean	SD	Mean	SD
Total	-	61.0	29.5	43.8	33.0	33.1	33.3
Stage	I	66.5	30.0	42.9	31.5	31.0	32.2
	II	48.3	25.3	45.7	37.7	37.9	36.9
Pain	no	61.7	28.7	46.2	29.0	33.1	32.7
	yes	60.4	30.9	42.1	36.2	33.2	34.6
Primary disease	osteoporosis	63.5	34.3	42.3	39.0	37.5	39.3
	malignoma	60.2	28.7	44.5	31.9	31.9	32.1
Risk evaluation	low risk	67.8	30.6	38.3	36.1	33.5	37.2
	high risk	57.5	29.0	46.5	31.8	33.0	32.1
Duration of intake	short	60.4	28.7	45.9	31.0	32.5	33.3
	long	61.3	30.8	42.3	35.1	33.6	34.2
Defect size	small	60.0	33.8	48.9	32.3	37.4	35.9
	medium	58.5	25.9	46.8	35.7	33.6	36.4
	large	63.6	30.7	37.4	32.8	29.4	30.9
Need for prostho-dontics	yes	75.1	25.0	47.4	35.3	41.0	36.9
	no	48.4	28.3	40.6	31.4	26.1	29.0

**Table 6 ijerph-19-11709-t006:** Mean VAS2-6 values for all parameters and assessment times (T0/T1/T2). SD: standard deviation.

Time of Assessment		T0		T1		T2	
Parameter	Groups	Mean	SD	Mean	SD	Mean	SD
Total	-	38.1	19.4	31.8	17.1	25.4	17.5
Stage	I	38.4	19.5	28.9	15.3	24.2	18.0
	II	37.5	20.1	38.4	20.0	28.3	16.6
Pain	no	38.7	20.5	35.4	15.4	28.0	19.3
	yes	37.6	19.1	29.2	18.2	23.6	16.3
Primary disease	osteoporosis	32.6	16.0	26.1	16.2	15.6	14.1
	malignoma	39.7	20.2	33.4	17.3	28.2	17.5
Risk evaluation	low risk	34.5	15.9	25.3	16.0	17.4	15.5
	high risk	39.9	21.0	35.0	17.1	29.4	17.3
Duration of intake	short	38.7	19.7	35.1	18.4	28.5	16.4
	long	37.6	19.6	29.4	16.2	23.3	18.3
Defect size	small	29.6	19.2	29.8	18.7	24.7	18.1
	medium	42.7	16.7	37.3	17.2	27.4	19.4
	large	41.2	20.6	29.0	16.1	24.5	16.7
Need for prostho-dontics	yes	44.1	20.6	35.4	16.7	29.8	18.7
	no	32.7	17.0	28.6	17.4	21.5	15.8

## Data Availability

The data presented in this study are available on request from the corresponding author.

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
