# Peer review of "Oral-Health-Related Quality of Life in Patients with Medication-Related Osteonecrosis of the Jaw: A Prospective Clinical Study"

_ijerph, 2022, doi:10.3390/ijerph191811709_

Round 1
Reviewer 1 Report
Dear Author (s)
1. There are grammatical errors. "stage I and II" to "stages I and II". etc.
2. There are several original articles related to the topic and even reviews and meta-analyses. What is the novelty?
3. Please add values of function, social, and psychological factors. Please see "https://bmcoralhealth.biomedcentral.com/articles/10.1186/s12903-021-01660-7/tables/1".
4. I think the number of cases is very low and this issue can affect on the correctness and reliability of the results.
Reviewer 2 Report
Were patients with cognitive impairments included? If not, it must be mentioned in the exclusion criteria.
Reviewer 3 Report
Review,
Thank you for referring me to evaluate this article.
The authors present a very valuable study evaluating the quality of life in patients who are affected by osteonecrosis of the mandible.
Although the study is very well structured and written after the evaluation, I can present the following aspects.
TITLE
The title - is suggestive for the analyzed study only that the article specifies that the data were collected at the time of TO and then after the surgical intervention line 22 and line 24, which does not mean drug treatment. I believe that the word medicinal should be eliminated or replaced with medical-surgical.
ABSTRACT
The abstract - it is well written, only that I have some suggestions:
- to be a Background phrase.
- to explain the purpose of the study.
- for material and method section, enter the number of evaluated subjects.
INTRODUCTION
The introduction is well structured, includes the defining elements of the conducted study and provides succinct data in the analyzed field supported by up-to-date bibliographic sources.
MATERIAL AND METHOD
- In the description of the protocol, the recording of the data regarding the date of obtaining the informed consent Line 100 and Line 405 is not highlighted correctly.
- How was the study group chosen? Is it representative? The authors present in the discussion chapter that one of the limitations is no. small number of subjects and refer to bibliographic source no. 33. It's just that this study was a pilot and had 30 patients. The other study to which I refer, respectively source 34, does not present the number of subjects.
- What is the Alpha coefficient and what is the strength of this study?
RESULT
They are presented succinctly and completely
DISCUSSIONS
The discussion chapter is presented with reference to the aspects addressed by the research theme and by the results obtained.
CONCLUSION
The conclusions are briefly presented.
REFERENCES
The bibliography is current. Most of the bibliographic references are from the last 5 years.
